# Optogenetic control of RhoA reveals zyxin-mediated elasticity of stress fibres

Patrick W. Oakes[1,2,3,4,5,*], Elizabeth Wagner[6,*], Christoph A. Brand[7], Dimitri Probst[7], Marco Linke[7], Ulrich S. Schwarz[7], Michael Glotzer[6] & Margaret L. Gardel[1,2,3]

Cytoskeletal mechanics regulates cell morphodynamics and many physiological processes. While contractility is known to be largely RhoA-dependent, the process by which localized biochemical signals are translated into cell-level responses is poorly understood. Here we combine optogenetic control of RhoA, live-cell imaging and traction force microscopy to investigate the dynamics of actomyosin-based force generation. Local activation of RhoA not only stimulates local recruitment of actin and myosin but also increased traction forces that rapidly propagate across the cell via stress fibres and drive increased actin flow. Surprisingly, this flow reverses direction when local RhoA activation stops. We identify zyxin as a regulator of stress fibre mechanics, as stress fibres are fluid-like without flow reversal in its absence. Using a physical model, we demonstrate that stress fibres behave elastic-like, even at timescales exceeding turnover of constituent proteins. Such molecular control of actin mechanics likely plays critical roles in regulating morphodynamic events.

[1] Institute for Biophysical Dynamics, University of Chicago, Chicago, Illinois 606037, USA. [2] James Franck Institute, University of Chicago, Chicago, Illinois 606037, USA. [3] Department of Physics, University of Chicago, Chicago, Illinois 606037, USA. [4] Department of Physics & Astronomy, University of Rochester, Rochester, New York 14627, USA. [5] Department of Biology, University of Rochester, Rochester, New York 14627, USA. [6] Department of Molecular Genetics and Cell Biology, University of Chicago, Chicago, Illinois 60637, USA. [7] Institute for Theoretical Physics and BioQuant, Heidelberg University, Heidelberg 69120, Germany. * These authors contributed equally to this work. Correspondence and requests for materials should be addressed to U.S.S. (email: schwarz@thphys.uni-heidelberg.de) or to M.G. (email: mglotzer@uchicago.edu) or to M.L.G. (email: gardel@uchicago.edu).

A diverse array of essential physiological processes, ranging from the subcellular to the multicellular, depend on the spatial and temporal regulation of contractile forces[1–4]. This regulation drives changes in cell shape[5,6] and mediates interactions with the extracellular environment[7,8]. Changes in contractility can furthermore alter gene expression[9] and impact development[10,11]. The molecular machinery required for generating contractile forces is well conserved and dominated by the actin cytoskeleton, myosin II activity and associated regulatory proteins[4,12,13]. Specifically, actin filaments dynamically organize into distinct contractile architectures, including the cortex and stress fibres[14,15]. Contractile forces are transmitted across the cell by actin arrays and ultimately to the extracellular matrix by focal adhesions[12,13,16,17].

The regulation of cellular force transmission is controlled by the mechanical properties of actomyosin assemblies. Cellular mechanics has been explored extensively both experimentally[14,18–20] and theoretically[21–23]. The current understanding is that, at timescales up to those of typical kinetic processes, the actin cytoskeleton behaves like an elastic solid. Such elasticity enables rapid force transmission across the cell and reversible deformations to preserve cytoskeletal architecture. In contrast, at longer timescales, it is thought that dynamic processes make the cytoskeleton behave predominantly like a viscous fluid, enabling cytoskeletal flows and remodelling. These dynamic processes, including exchange of proteins from the cytosol, are typically on the order of tens of seconds in structures like the cortex[24,25] and on the order of a minute in stress fibres[26]. The molecular regulation underlying the competition between elastic and viscous processes in cells is not well understood.

Cellular contractility is largely controlled by the activity of the small GTPase RhoA[27,28], which in adherent cells is preferentially active at the cell periphery[29,30]. RhoA regulates contractility through the promotion of actin polymerization and myosin light chain (MLC) phosphorylation via the downstream effectors Diaphanous-related formins and Rho-associated kinase (ROCK), respectively. RhoA activity is required for stress fibres and focal adhesions[27,28]. Little, however, is known about how small changes in activity can regulate cell contractility, actin architecture and adhesion.

Optogenetics has been used before to control cell migration and cell forces[31–34]. Here we have used an optogenetic probe to locally activate RhoA in adherent fibroblasts. Plasma membrane recruitment of the RhoA-specific guanine exchange factor (GEF) LARG induces local RhoA activation[6,35–37]. Local activation of RhoA leads to an increase in actin polymerization and myosin activity in the region of activation, but it does not stimulate de novo stress fibre formation or changes in focal adhesion morphology. We find that exogenous RhoA activation leads to an immediate increase in both the local and global contractility of the cell, followed by a rapid relaxation after GEF recruitment is stopped. The local increase in stress fibre contractility drives an actomyosin flow towards regions of increased RhoA activity. Surprisingly, these flows reverse direction as soon as GEF recruitment ceases. Using physical modelling, we show this behaviour is consistent with stress fibres behaving as predominately elastic-like over timescales much longer than minutes. We find that zyxin is necessary for this elasticity; in its absence, stress fibres become predominantly fluid-like even at second timescales. These results suggest that stress fibre mechanics are sensitive to small changes in composition, which has significant implications for regulation of force transmission and cytoskeletal organization.

## Results
### Spatiotemporal control of RhoA and its downstream effectors.
To spatially and temporally control contractility in adherent cells, we adapted a previously established optogenetic probe[6,35] to act

on the RhoA signalling pathway (Fig. 1a). During stimulation by blue light, a cytosolic fusion protein, photo-recruitable GEF (prGEF), consisting of tandem PDZ domains fused to the DH domain of the RhoA-specific GEF LARG[6], is recruited to the plasma membrane where it activates RhoA (Fig. 1b). To illustrate the local recruitment of prGEF, we tagged it with the fluorophore mCherry and imaged an NIH 3T3 fibroblast expressing the constructs on a glass coverslip (Fig. 1c). A digital micromirror device was used to spatially control the illumination of the blue-activating light (orange box, Fig. 1c) and was pulsed before each image acquisition during the recruitment period. Recruitment of the prGEF to the activation region was rapid and reversible upon extinguishing the stimulating blue light (Fig. 1c,d and Supplementary Movie 1).

To investigate whether recruitment of the prGEF resulted in activation of proteins downstream of RhoA, we tracked the dynamics of actin and MLC during recruitment (Fig. 1e–g). Both actin and myosin accumulated in the activation regions, resulting in an increase in fluorescence intensity during the 15 min activation period (Fig. 1e–g and Supplementary Movies 2 and 3). Interestingly, local activation of RhoA did not lead to de novo stress fibre assembly in the activation region (Fig. 1e and Supplementary Movie 2). At the end of the activation period, fluorescence intensities of both actin and myosin returned to baseline levels. These results indicate that exogenous RhoA activation via LARG recruitment is not sufficient to maintain elevated RhoA activity and the concomitant increases in local actin and myosin concentrations. To confirm that local activation of RhoA was acting on actin and myosin through its downstream effectors, formin and ROCK, the experiments were repeated in the presence of either SMIFH2, a pan formin inhibitor[38], or Y-27632, a ROCK inhibitor (Fig. 1h). Local recruitment of actin and MLC were significantly inhibited by the presence of SMIFH2 and Y-27632, respectively. These results illustrate that RhoA activity and recruitment of its downstream effectors can be spatially and temporally controlled via light.

### Adhesion morphology is unperturbed by local RhoA activity.
Previous work has suggested that focal adhesion formation and maturation are tension-dependent processes driven by increased RhoA activity at adhesion sites[27,28,39]. To test these hypotheses, we examined how local RhoA activation affected traction forces and focal adhesions. Cells expressing mCherry-vinculin, a marker of focal adhesions, were plated on polyacrylamide gels coated with fibronectin and traction stresses were measured via traction force microscopy (Fig. 2a,b and Supplementary Movie 4)[40,41]. During local activation of RhoA, traction stresses increased at focal adhesion sites, on a similar timescale to that of myosin localization (Fig. 1g). Despite the increased force, the total number of adhesions remained essentially constant (Fig. 2c). Individual adhesion morphology and vinculin intensity were also unaffected, despite large increases in stress at a majority of previously established adhesion sites during local RhoA activation (Fig. 2d and Supplementary Fig. 1). Too much illumination, however, results in adhesion failure and detachment from the substrate (Supplementary Fig. 2). As the adhesions and stress fibres are fully formed already in these cells, the lack of change in adhesion morphology is consistent with previous results showing that adhesion size and tension are only correlated during the initial growth phase of the adhesion[42] and that adhesion maturation is driven by proximal actin stress fibre assembly[43].

To determine the effect of local RhoA activation on the overall contractility of the cell, we used traction force microscopy to measure the total strain energy, which includes both near and far field deformations and reflects the total amount of mechanical

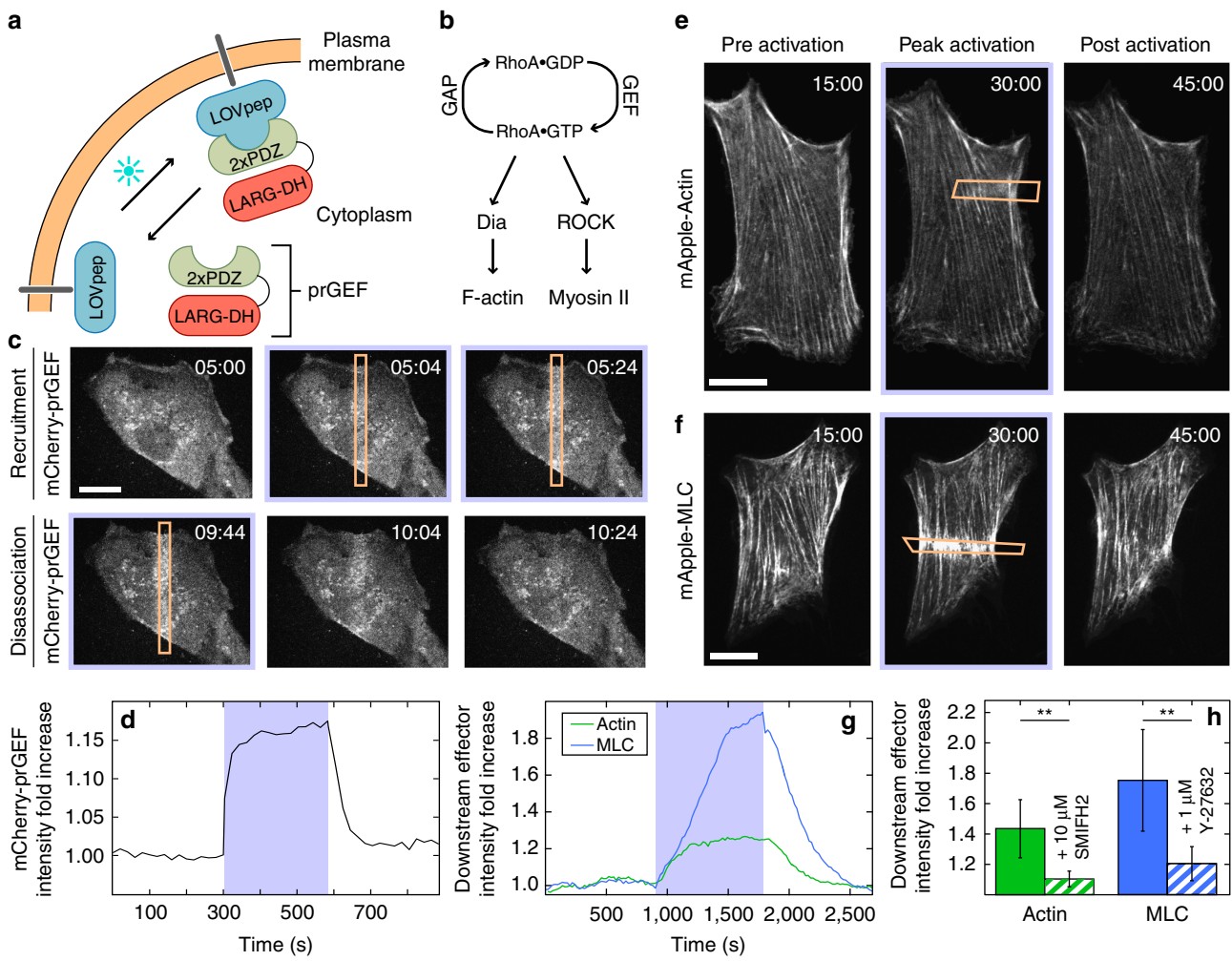

**Figure 1 | RhoA activity can be spatiotemporally controlled via an optogenetic probe.** (**a**) Spatiotemporal control of RhoA activity is achieved using an optogenetic probe to recruit the RhoA-specific GEF LARG to the plasma membrane. A LOVpep molecule is anchored to the membrane via fusion to the transmembrane protein Stargazin, while a protein consisting of tandem PDZ domains fused to the DH domain of LARG (prGEF) is distributed throughout the cytosol. Upon stimulation with 405 nm light, the LOVpep undergoes a conformational change exposing a high-affinity binding site, which drives the prGEF to the membrane where it can activate RhoA. When the activating light is removed, the LOVpep undergoes a thermodynamically driven refolding halting further recruitment of prGEF. (**b**) The RhoA signalling pathway. RhoA•GTP activates both Diaphanous-related formins and ROCK, which in turn promote actin polymerization and myosin II activation, respectively. (**c**) Representative images of an NIH 3T3 fibroblast expressing an mCherry-tagged version of prGEF. Upon local activation (top row—orange box), mCherry-prGEF rapidly accumulates in the activation region. Removal of the activating light (bottom row) results in the accumulated mCherry-prGEF dispersing back into the cytosol. (**d**) Quantification of the local intensity increase of mCherry-prGEF in the activation region of the cell shown in panel **c**. The activation period is indicated by a blue background. (**e,f**) Representative images of cells expressing either mApple-Actin (**e**) or mApple-MLC (**f**) before activation, following 15 min of activation in the region indicated and following 15 min of relaxation. Both actin and myosin exhibit increases in intensity in the local region of activation. (**g**) Quantification of the local intensity increase of actin and myosin from the cells in panels **e,f**. Both signals begin increasing immediately upon RhoA activation and dissipate as soon as the activating light is switched off. (**h**) Mean maximum intensity fold increase of actin (n = 6) or myosin (n = 11) in regions of activation in control cells or cells treated with 10 μM SMIFH2 (pan formin inhibitor; n = 7) or 1 μM Y-27632 (ROCK inhibitor; n = 6). Inhibition of either Dia or ROCK results in reduced average increases in local intensity during RhoA activation. Error bars represent s.d. **P < 0.01, two-sample Student's t-test. Time is min:s. Scale bars are 15 μm.

work done by the cell on its environment[44]. Activation induced a rapid increase in both traction stresses and strain energy (Fig. 2b,c). At the end of the activation period, the strain energy decreased to its original baseline value (Fig. 2c). Interestingly, traction stresses were mostly seen to increase at the cell periphery, where traction stresses were already established, and in areas immediately adjacent to the activation region. No change was seen in the activation area itself (Fig. 2b). This suggests that locally generated forces balance within the activation region and only unbalanced forces at the edge of this region are turned into productive traction forces. Thus a local

increase of tension leads to globally distributed traction forces at pre-existing focal adhesions.

**Cells maintain a contractile set point.** That cells return to a similar baseline contractility following a period of exogenous RhoA activation is consistent with previously established ideas of tensional homeostasis[44–47]. To explicitly probe this behaviour, we performed a series of local activations of different sizes on a single cell (Fig. 2e and Supplementary Movie 5). After measuring the strain energy at an initial steady state, a cell was exposed to

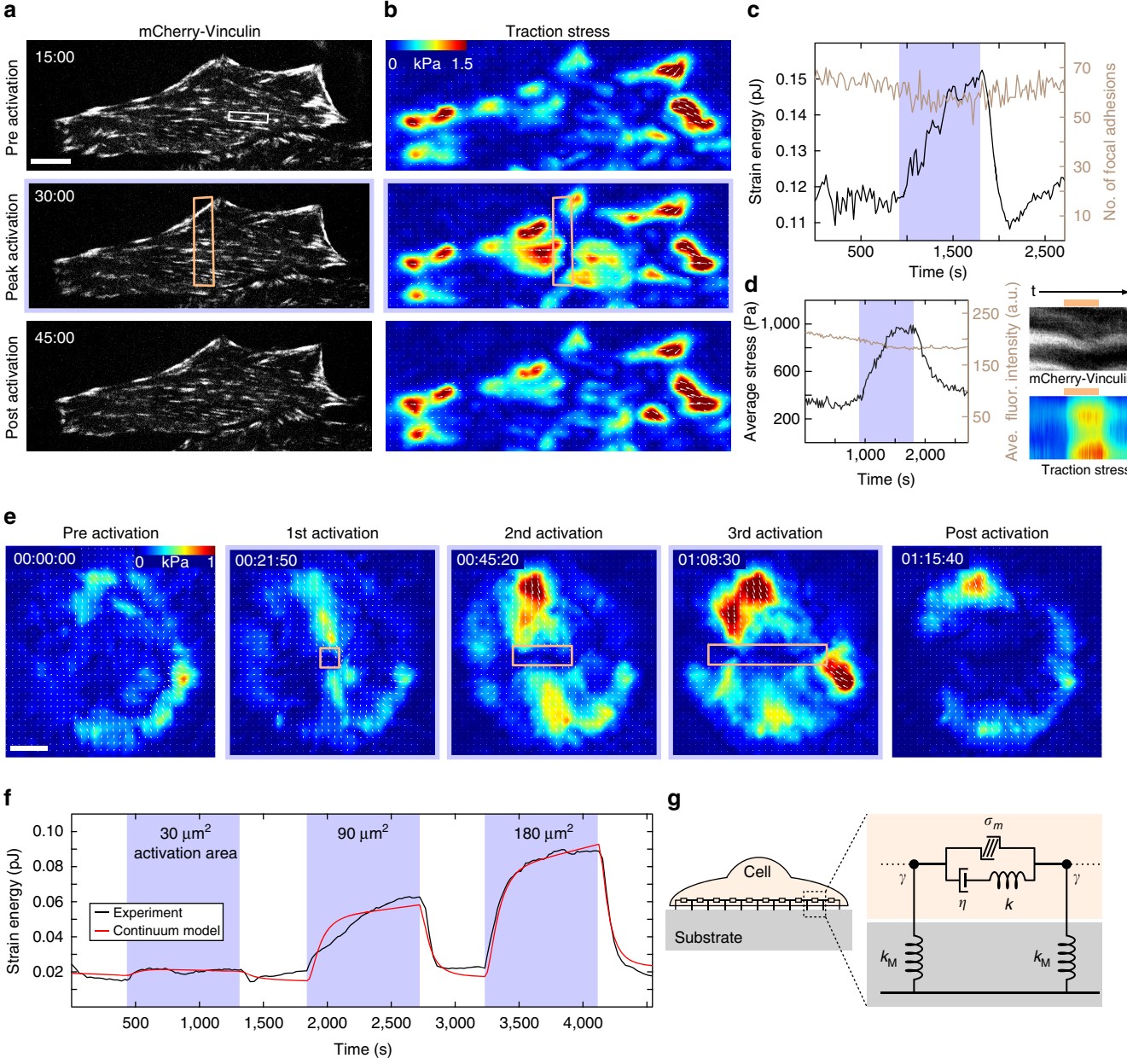

**Figure 2 | Increased RhoA activity leads to increased cell contractility.** (**a**) A cell expressing mCherry-Vinculin is shown before activation, after 15 min of activation and after 15 min of relaxation. The activation region is indicated by the orange box. (**b**) The corresponding traction stress maps for the time series shown in panel **a**. (**c**) A plot showing the strain energy versus time and the number of focal adhesions versus time, with the activation period indicated by the blue background. The strain energy begins to increase immediately upon activation and begins to relax as soon as the activating light is removed. In contrast, the number of focal adhesions remains relatively constant and does not respond to the local increase in RhoA activity. (**d**) The average stress and fluorescence intensity of a representative adhesion marked by the white box in panel **a**. Kymographs were generated by drawing a line along the long axis of the adhesion. The activation period is indicated by the orange bar above the kymograph. (**e**) A sequence of traction maps from a cell exposed to a series of activations in regions of different size. Time is h:min:s. (**f**) A plot of the experimental (black line) and theoretical (red line) strain energy versus time for the cell shown in panel **e**. The contractile response of the cell is proportional to the size of the activation region and retreats to a baseline value following each activation period. (**g**) A cartoon of the continuum model used to describe the cell in panel **e**. The model consists of a contractile element ($\sigma_{\mathrm{m}}$) in parallel to a viscous ($\eta$) and an elastic element ($k$) in series, connected via a frictional elements ($\gamma$) to an elastic substrate ($k_{\mathrm{M}}$). Scale bars are 10 μm.

three 15 min periods of local RhoA activation of increasing size with relaxation periods between each activation (Fig. 2e,f). The strain energy increased concomitant with the size of the activation region. During activation, both local stresses immediately surrounding the activation region, and long-range stresses at the cell periphery could be seen to increase (Fig. 2e). Following each activation, the strain energy returned to the initial baseline level (Fig. 2f).

To elucidate the underlying mechanical principles, we built a physical model that would capture this physical response (see Supplementary Note for details). We constructed a model of the cell as a continuum of contractile elements in series, each in parallel with one elastic and one viscous element in series (Fig. 2g). Such a model is known as an 'active Maxwell fluid' and is used here because we are interested in long timescales, when the system is expected to flow[21,25,48]. Contractility was assumed

to increase with an exponentially plateauing ramp in the activated region, consistent with the observed accumulation profiles for actin and myosin (Fig. 1g), and the substrate was represented as an elastic spring coupled to the cell by a friction element. The model parameters for the elastic modulus, viscosity, friction and contractility were found by fitting the model to the strain energy data, while the value of the substrate stiffness was fixed. This procedure resulted in a curve in good agreement with the experimental data (Fig. 2f).

We find that both the viscous and elastic elements are necessary to accurately capture the behaviour of the system over the whole time course of ramping, plateau and relaxation. The ratio of viscosity to elasticity defines a viscoelastic relaxation time of approximately 60 min; this timescale determines the transition from when the cytoskeleton behaves predominately elastic-like ($<60$ min) to predominately fluid-like ($>60$ min). Our results thus indicate that stress fibres are predominantly elastic on the scale of tens of minutes, despite proteins within the stress fibre turning over on timescales of tens of seconds[15]. This strong elastic behaviour is also consistent with the immediate increase in cell traction stresses at the cell periphery (Fig. 2b,e) upon local activation of RhoA in the centre of the cell.

**Stress fibres contract independent of the background network.** To investigate the cytoskeletal architectures that give rise to this strong contractility, we tracked myosin dynamics during local RhoA activation. In the steady state, as new actomyosin is polymerized and incorporated into stress fibres, there is a retrograde flow of actomyosin from the periphery towards the cell centre[15,17]. Using particle image velocimetry[49,50], we measured both the local direction and magnitude of myosin flow. We found that myosin flow rates along the stress fibre increased as myosin accumulated in the activation region creating a local contraction and that this flow was directed along the orientation of the stress fibres spanning the activation region (Fig. 3a,b, Supplementary Fig. 3 and Supplementary Movie 6). The flow direction was independent of the activation region geometry, with the direction always being determined by the stress fibre orientation (Supplementary Movie 7).

The cytoskeleton of a strongly adherent cell is typically thought to be a two-dimensional (2D) material comprised of stress fibres embedded in an isotropic actin network[41,51,52]. Since flows induced by local RhoA activation appear to track the orientation of the stress fibres (Fig. 3a,b and Supplementary Fig. 2), we sought to address the relative contractile contributions of the stress fibres and the actin networks. We therefore built a 2D discrete model analogous to the one-dimensional (1D) continuum model described above (Fig. 3c; Supplementary Note). The model consists of a triangular mesh with the contractile elements again connected in parallel to the viscous and elastic elements, with lines of increased contractility representing the stress fibres. Using a simple rectangular cell, we first verified that, without stress fibres, this model recapitulates the results from the 1D continuum model (Fig. 3d). Similar to the 1D model above, the contractile components in a region in the centre of the cell were slowly increased with an exponentially plateauing ramp. The parameters were then adjusted so that the model both qualitatively and quantitatively recapitulated the expected flow patterns of the 1D continuum model (Fig. 3d).

To explore the relative contributions of the background mesh and the stress fibres, we considered two test cases: (1) If both the mesh and the stress fibres contained contractile elements, the stress fibres pinched inward transverse to their orientation during local activation (Fig. 3e); and (2) when contractile elements were only included in the stress fibres (as depicted in Fig. 3c), the

cytoskeletal flow was restricted to directions along the stress fibre (Fig. 3f), consistent with our experimental results (Fig. 3a,b, Supplementary Fig. 3 and Supplementary Movies 6 and 7). Since transverse deformations were never seen in experiments, it is clear that the stress fibres must be the predominant contractile elements observable at this resolution that respond to local RhoA-induced contractions. Furthermore, this result illustrates that it is appropriate to think of a stress fibre as a 1D contractile element with viscous and elastic components embedded in a passive viscoelastic network, which is consistent with recently published image-based observations[26].

**Stress fibres flow due to local RhoA induced strain.** Having identified the stress fibre as the main contractile unit responding to exogenous RhoA activation, we next sought to address whether stress fibres undergo deformation during contraction. Since stress fibres can be considered as 1D structures, we analysed myosin flow along the fibre using kymographs. A kymograph drawn along a single stress fibre illustrates that myosin puncta flowed from both ends towards the activation regions when RhoA was activated locally (Fig. 4a,b, Supplementary Fig. 5). Similarly, a kymograph drawn by projecting the flow speed along the stress fibre from the velocity field created by our particle image velocimetry analysis illustrates even more clearly how cytoskeletal flow was perturbed by local RhoA activation. Flow of myosin from both ends of the stress fibre reoriented towards the recruitment regions and increased from $\sim 1$ nm s$^{-1}$ on average to $>3$ nm s$^{-1}$ during activation (Fig. 4c and Supplementary Fig. 5). Strikingly, the flow was also seen to reverse direction, flowing away from the recruitment region and towards the cell periphery, during the relaxation period following the local activation (Fig. 4b,c). This flow reversal is reminiscent of the restoring force in elastic objects that restores its original shape after removal of external force (for example, recoil of an elastic band after stretch). In active systems, flow reversal could also arise from spatial variations in tension. Specifically, this would either require the myosin stress within the activation region to fall below its preactivation level or increased myosin activity distal to the activation site. However, we do not observe such changes in actomyosin density, indicating that a passive elastic-like element may be sufficient in describing this recoil.

We next developed a protocol to measure the magnitude of the stress fibre displacement during these periods of contraction and relaxation (Fig. 4d). The displacement in a given fibre was determined by measuring the relative position of puncta along the fibre following 15 min of local RhoA activation and 15 min after it ceased. During contraction, puncta on either side of the activation region contracted on average $\sim 3$ µm from their original position before relaxing back to $\sim 1$ µm from their original position (Fig. 4d). The relaxation response across many stress fibres from multiple cells could be further clustered into two groups, one which exhibited strong reversal ($\sim 80\%$ of the original position) and one which exhibited little to no reversal ($\sim 25\%$ of the original position) (Fig. 4d).

To determine whether stress fibres were stretching due to the local contraction, we used cells expressing mApple-α-actinin, an actin crosslinker that localizes to well-defined puncta on stress fibres (Fig. 4e and Supplementary Movie 8). We created kymographs of α-actinin flow during local activation of RhoA and tracked paths of individual puncta (Fig. 4f). The velocity of individual puncta was determined from the slope of the tracks in the kymograph and plotted as a function of distance from the activation zone (Fig. 4g). Puncta along the stress fibre moved at similar speeds, indicating that, in general, the stress fibre was translating as a rigid rod during the local contraction (Fig. 4g).

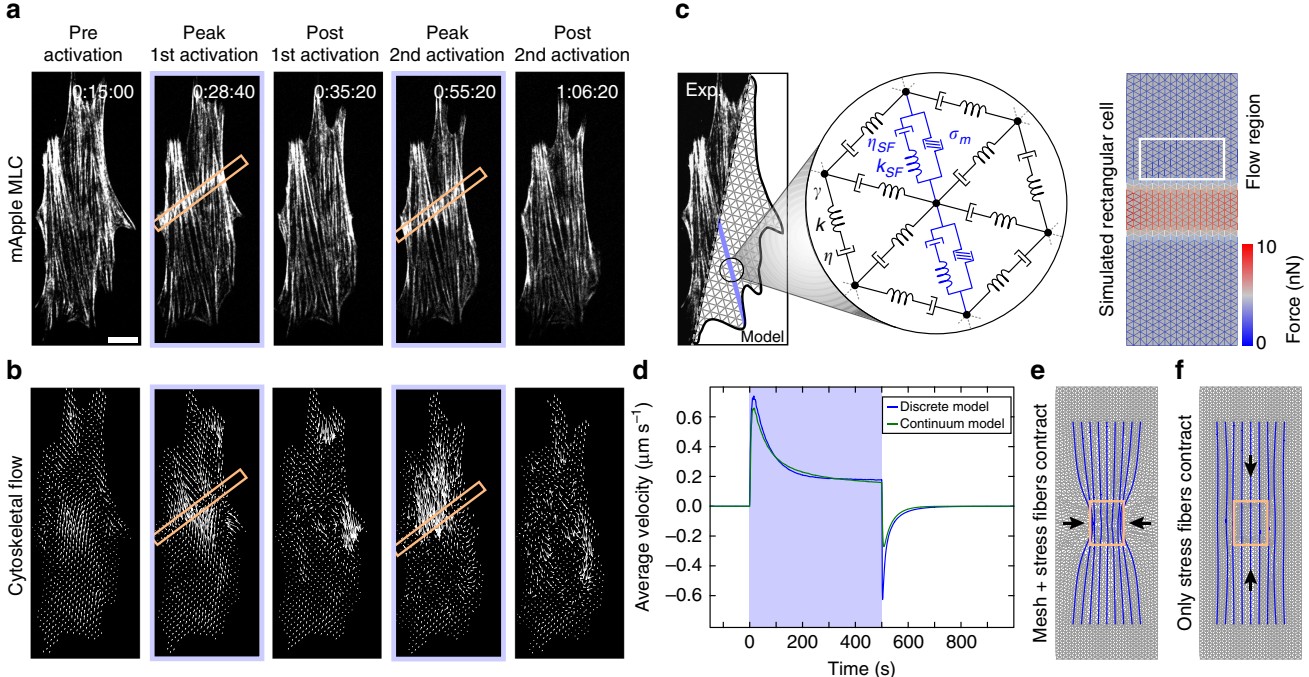

**Figure 3 | Stress fibers direct contractile flow.** (**a**) Fluorescence time series of a cell expressing mApple-MLC shown prior and after two separate periods of activation (orange boxes). (**b**) Flow fields of myosin calculated from the images in panel **a**. Flow is always directed along the direction of the stress fibers. (**c**) A two-dimensional model of the cell was created using a triangular mesh of viscoelastic cables ($k,\eta$) connected at vertices viscously coupled ($\gamma$) to the environment. Stress fibres (blue line) consisting of contractile ($\sigma_m$), viscous ($\eta_{SF}$) and elastic elements ($k_{SF}$) were embedded in the network. Using a simplified rectangular cell with this network, local RhoA activation could be simulated by activating force dipoles in network links in a region in the centre of the cell. (**d**) To calibrate the 2D discrete model, the average flow (white box in panel **c**), was measured and compared to the 1D continuum model presented above. (**e,f**) The 2D discrete model was used to explore two contractile scenarios: (**e**) contractile stress fibres (blue) embedded in a contractile mesh (grey); and (**f**) contractile stress fibres embedded in a non-contractile background. If both the stress fibres and mesh are contractile, a transverse contraction pinches together the stress fibres. If only the stress fibres contract, the flow profile is restricted to the orientation of the fibres, mimicking the experimental results. Scale bar is 10 μm.

Where present, changes in velocity between neighbouring puncta were abrupt (blue arrow, Fig. 4g), suggesting points of structural failure along a fibre. These results indicate that the strain induced in the stress fibre is restricted to the local contraction in the activation region and discrete sites of extension in regions outside the activation region.

By fitting the experimental kymographs to both our 1D continuum and 2D discrete models, we show that similar flow patterns emerge naturally from the mechanics of the system (Fig. 4h). The high elasticity of the stress fibre, specifically the ratio of elasticity to viscosity, is sufficient to recapitulate the flow profiles that were seen during both RhoA activation and relaxation. Furthermore, the parameters found from the kymograph fitting process were consistent with the parameter values found when fitting the strain energy (Supplementary Table 2).

**Zyxin is recruited to sites of strain on stress fibres**. In order to probe the underlying molecular basis of this elasticity, we sought to identify stress fibre-associated proteins that could contribute to the recoil behaviour. Zyxin has been previously established as a mechanosensitive protein that dynamically localizes to sites of strain along stress fibres[53,54], in addition to focal adhesions[55]. Using cells expressing mCherry-zyxin, we monitored zyxin localization during RhoA activation (Fig. 5a and Supplementary Movie 9). Zyxin recruitment was consistently observed in a small population of focal adhesions outside of the local activation

region (Fig. 5b). Surprisingly, we found that zyxin also accumulated along stress fibres in the region of local activation (Fig. 5c,d). Given the myosin accumulation and direction of flow, this suggests that zyxin might be recruited to both sites of compression and extension. Paxillin, another mechanosensitive LIM domain protein that responds to stress[56,57], behaved similarly to zyxin (Supplementary Fig. 4).

**Zyxin is required for stress fibres to behave elastically**. To further explore the role of zyxin in stress fibre mechanical behaviour, we used mouse embryonic fibroblast cells derived from zyxin$^{(-/-)}$ mice[58]. Despite the loss of zyxin, these cells form actin stress fibres and focal adhesions and are highly contractile[59]. When we locally activated RhoA in the zyxin$^{(-/-)}$ cells, myosin accumulated in the activation region (Fig. 5e and Supplementary Movie 10). This accumulation drove a contractile flow into the local activation area that was indistinguishable from wild-type cells, indicating that loss of zyxin did not impede myosin activity. Upon stopping the GEF recruitment in zyxin$^{(-/-)}$ cells, cytoskeletal flow returned to preactivation rates, consistent with the reduced local contraction, but did not reverse direction (Fig. 5e,f,i, Supplementary Fig. 5, and Supplementary Movie 10). Expression of enhanced green fluorescent protein (EGFP)-zyxin in this cell line restored the flow reversal (Fig. 5g–i, Supplementary Fig. 5 and Supplementary Movie 11). Together these results indicate that zyxin is required for the flow reversal occurring after local RhoA activation ends.

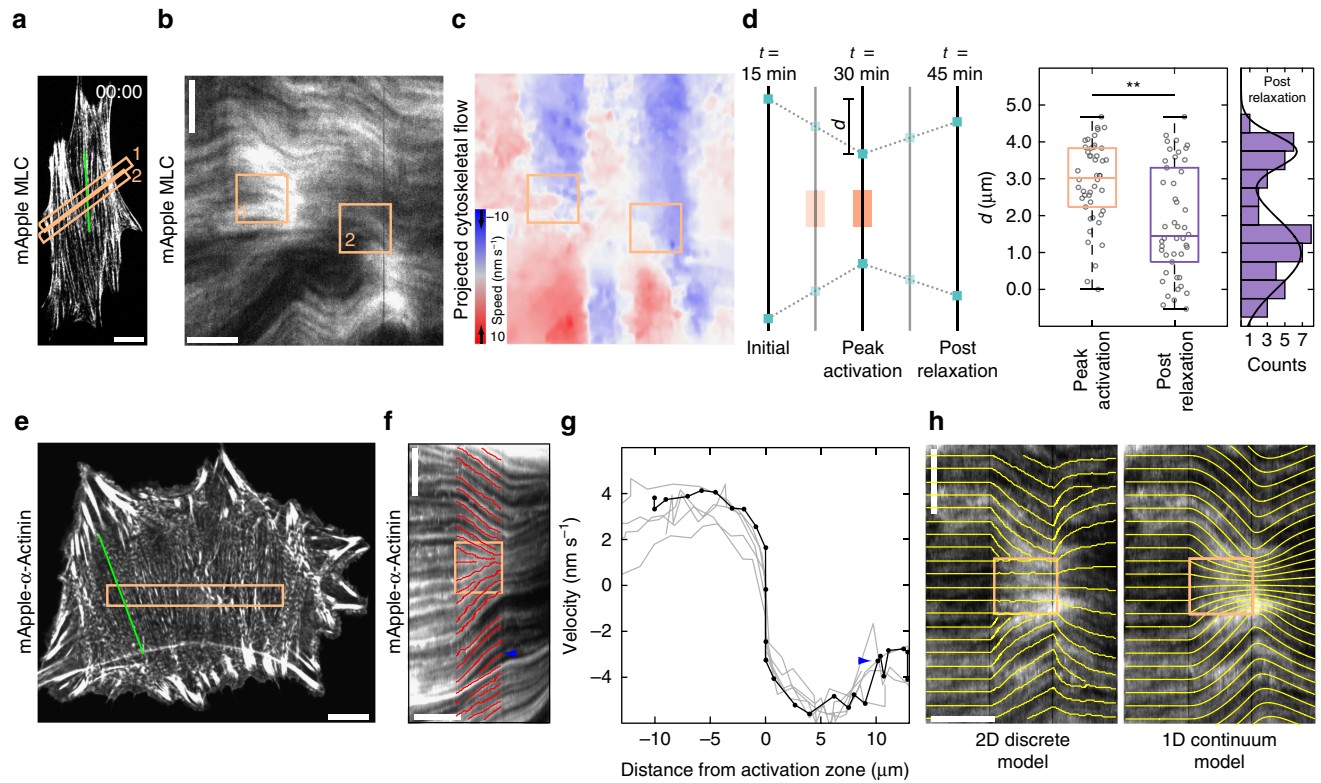

**Figure 4 | Stress fibres behave elastically.** (**a**) Image showing a cell labelled with mApple-MLC. The activation regions are indicated by the orange boxes. (**b**) A kymograph drawn along the stress fibre (green line in panel **a**). During activation periods, myosin flows towards the activation regions. (**c**) A kymograph created from the same region as panel **b** using the flow maps determined previously. Flow was projected onto the stress fibre and colour coded to indicate speed and direction. This flow map illustrates that, during relaxation periods, myosin flow reverses direction away from the activation periods. (**d**) A quantification of displacement of stress fibres during contraction and relaxation. Puncta ∼5 μm from the activation zone were tracked and measured following 15 min of activation and again following 15 min of relaxation ($n = 41$ from 4 cells). Puncta translated about 3 μm from their original position and relaxed to ∼1 μm from their original position elastically. The relaxation response could be further broken into two groups, one with a strong reversal (∼80% of their original position) and one with a weak reversal (∼25% of their original position). (**e**) A cell transfected with mApple-α-actinin. The activation area is indicated by the orange box. (**f**) A kymograph drawn along the direction indicated in panel **e**, overlain with tracks of the individual α-actinin puncta during activation. (**g**) The velocity of individual puncta along the stress fibre is measured from the slope of the tracks and plotted against the distance from the activation region. Adjacent puncta all move at approximately the same speed. Sudden changes in velocity (blue arrowhead) correlate with what appear to be site of mechanical failure along the stress fibre and the appearance of new puncta. The black line represents the stress fibre from panel **f**, while the grey lines are other stress fibres from the same cell. (**h**) A representative kymograph is fit to both the 1D continuum and 2D discrete models. Both models are able to recapitulate the flow patterns seen experimentally. Scale bars are 10 μm in panels **a,e**. Horizontal scale bars are 15 min and vertical scale bars are 5 μm in panels **b,f,h**. **$P < 0.01$, two-sample Student's $t$-test.

Using the kymographs produced in the zyin$^{(-/-)}$ and zyxin$^{(-/-)}$+EGFP-zyxin cells, we again fit the data to our mechanical model (Fig. 5j–l, Supplementary Fig. 6). For the zyxin$^{(-/-)}$ cells, we found the viscoelastic relaxation time reduced to 1 s, indicating that the stress fibres are predominantly fluid-like at all physiological timescales. Rescue of the zyxin$^{(-/-)}$ cells with EGFP-zyxin resulted in parameter fits that were consistent with the NIH 3T3 fibroblast data. Zyxin is thus important for maintaining the qualitative mechanical response of stress fibres, ensuring that they are predominantly elastic at ∼1 h timescales.

## Discussion

This study demonstrates that the mechanical behaviour of adherent cells is strongly shaped by stress fibres and their ability for rapid force transmission even in the face of molecular turnover and flow. Using an optogenetic probe to locally activate RhoA via recruitment of the DH domain of LARG, a RhoA-specific GEF, we find that we can stimulate a local contraction in stress fibres due to an increased accumulation of actin and myosin in the activation area (Fig. 6, 1). This local contraction causes a tension gradient at the boundaries of the activation region and a flow towards it (Fig. 6, 2). The flow of myosin and α-actinin increases the strain both on the interface coupling the stress fibre to the adhesion and in the activation region, leading to recruitment of the mechanosensitive protein zyxin (Fig. 6, 3). When local activation of RhoA is stopped, the system relaxes to the preactivation state, mainly driven by elastic energy accumulated in the strained regions, and results in a transient cytoskeletal flow of material away from the local activation region (Fig. 6, 4).

This elastic behaviour is dependent on zyxin. Previous reports have shown that zyxin localizes along the stress fibre at the interface of the adhesion[60,61]. This positioning suggests that previously reported zyxin-mediated stress fibre repair mechanisms[56,59] are also occurring at the adhesion interface as actin is assembled and is incorporated into the stress fibre while under tension. The localization of zyxin to potential sites of compression, however, is novel. While it is known that the LIM

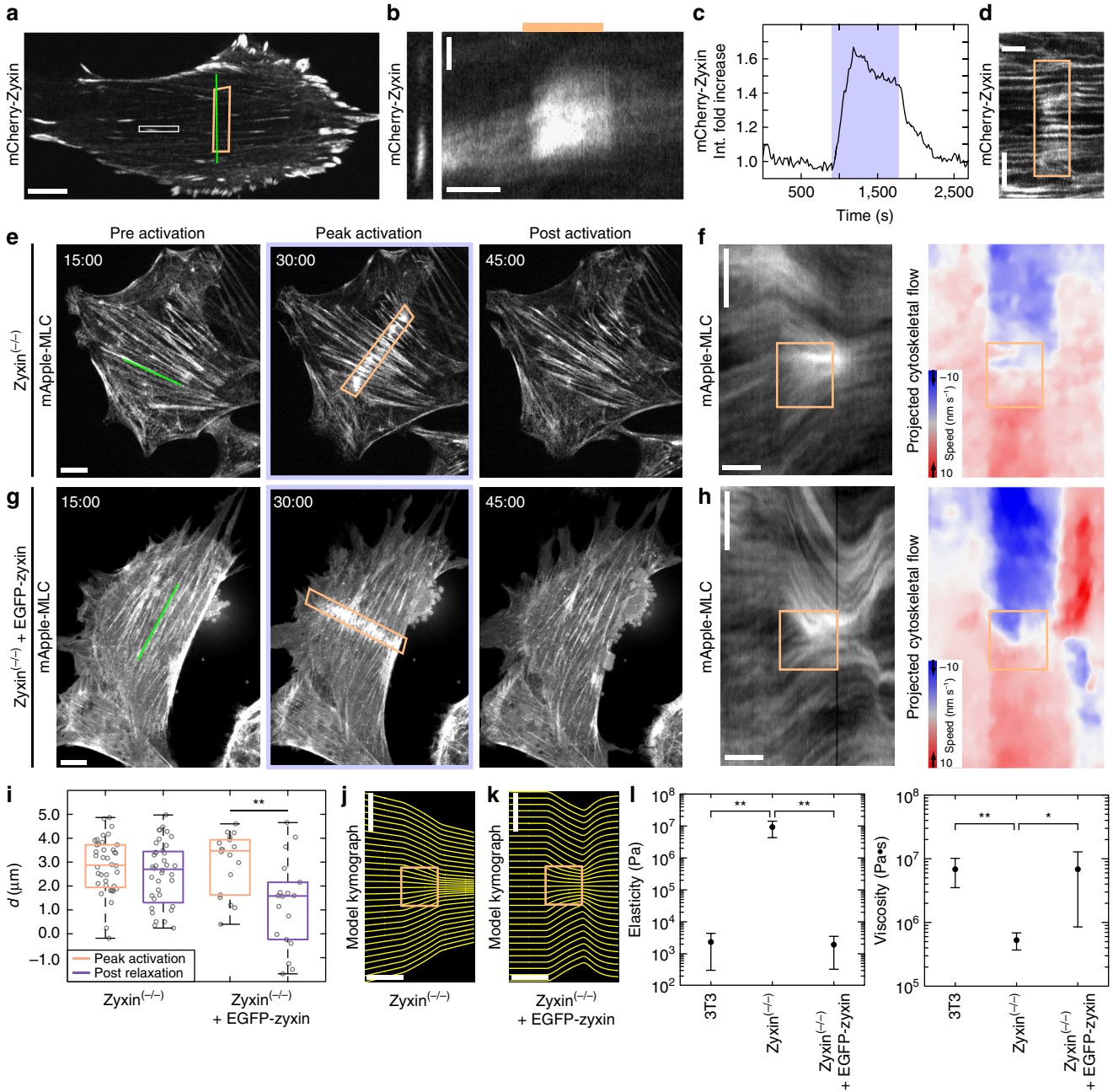

**Figure 5 | Zyxin accumulates at sites of strain on stress fibres during local RhoA activation.** (**a**) A NIH 3T3 expressing mCherry-zyxin. The activation region is indicated by the orange box. (**b**) A kymograph of a representative adhesion marked by the white box in panel **a**. (**c**) The average intensity of zyxin in the activation region. (**d**) A kymograph illustrating the local zyxin accumulation during activation along the green line shown in panel **a**. (**e**) A zyxin$^{(-/-)}$ MEF expressing mApple-MLC before activation, at peak activation and following relaxation. Myosin accumulates as in 3T3s. (**f**) A kymograph of myosin intensity and flow speed drawn along the green line indicated in panel **e**. Zyxin$^{(-/-)}$ MEFS exhibit little to no elastic flow reversal. (**g**) A zyxin$^{(-/-)}$ MEF rescued with EGFP-zyxin and expressing mApple-MLC, during an activation sequence. (**h**) A kymograph of myosin and flow illustrating a strong elastic flow reversal along the line drawn in panel **g**. (**i**) Displacement analysis of zyxin$^{(-/-)}$ ($n = 40$ from 3 cells) and zyxin$^{(-/-)}$ + EGFP-zyxin MEFs ($n = 18$ from 3 cells). Without zyxin, cells do not exhibit an elastic response. (**j**) A kymograph representing the average fit of the continuum model to the zyxin$^{(-/-)}$ data. (**k**) A kymograph representing the average fit of the continuum model to the zyxin$^{(-/-)}$ + EGFP-zyxin data. (**l**) The elastic (*E*) and viscous (*η*) parameters found from fitting the experimental kymographs to the continuum model ($n = 5$ from 3 cells for condition). Without zyxin, the elasticity increases and the viscosity decreases. Scale bars are 10 μm in panels **a,e,g**. Horizontal scale bars are 10 min in panels **b,d,f,h,j,k**. Vertical scale bars are 2 μm in panel **b** and 5 μm in panels **d,f,h,j,k**. *$P < 0.05$; **$P < 0.01$, two-sample Student's *t*-test.

domain of zyxin is sufficient for localization[62], the exact mechanism through which zyxin recognizes sites of strain remains unknown.

These data further illustrate that RhoA activity and its downstream effectors are tightly regulated by the cell. We see no evidence that RhoA activation alone leads to *de novo* stress fibre formation or adhesion maturation. Instead these processes likely result from concurrent changes in cytoskeletal architecture[43,63]. More interestingly, the data suggest cells actively regulate total RhoA activity to maintain a constant

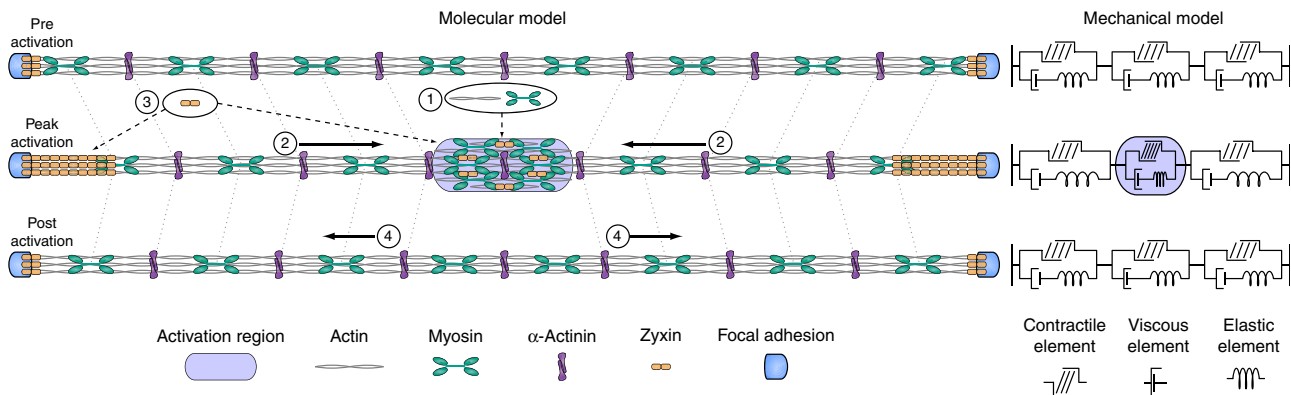

**Figure 6 | Molecular and mechanical models of local RhoA activation in stress fibres.** Molecular model: (1) Local recruitment of prGEF leads to activation of RhoA and accumulation of actin and myosin. The local increase in actin and myosin in turn stimulates a local contraction in the stress fibre. (2) The increased local contractility induces a flow of myosin and α-actinin along the stress fibre towards the activation region. (3) Increased flow induces higher strain at both the interface coupling the stress fibre to the focal adhesion and the activation region, resulting in recruitment of the mechanosensitive protein zyxin. (4) When local activation of RhoA stops, the flow reverses direction as the stress fibre relaxes elastically. Mechanical model: The stress fibre is represented as a contractile element in parallel to viscous and elastic elements. Upon local activation of RhoA, the contractile element inside the activation region is compressed, leading to extension of both the viscous and elastic elements outside the region. As local activation stops, the energy stored in the elastic element allows it to relax back to its original state, while the energy in the viscous element is lost.

homeostasis[44–47]. Specifically, the relaxation kinetics of the downstream effectors match the kinetics of the optogenetic probe[35], thus indicating that there is no positive feedback loop whereby production of RhoA●GTP alone is sufficient to promote further activation of RhoA. To sustain a given contractile state, therefore, the cell must actively regulate and maintain a specific RhoA●GTP concentration.

By using an optogenetic approach to perturb the local mechanical balance within the cell, we were able to probe the material properties of the cytoskeleton in ways previously inaccessible. It is instructive to consider our results within the context of other approaches that probe stress fibre mechanics. One such method is the cyclic stretch of an elastic substrate, in which Rho activation is involved in reorienting the cell and its stress fibres away from the direction of stretch[64]. It has been argued that the reorientation response might result from a homeostatic set point, that is either passive[65] or active[66] in nature. Our optogenetic experiments also reveal a homeostatic mechanism, which depends on the repair protein zyxin. This suggests that the homeostasis we describe has evolved to be actively maintained and is not simply an elastic response of a passive material. Another interesting comparison can be made with severing of stress fibres via laser ablation[53,67–69]. In severing experiments, the mechanical stability of the fibre is compromised while myosin activity on the remaining portion remains unchanged, thus driving a large contraction over seconds. By contrast, we probe the response to small perturbations in tension along an intact stress fibre over tens of minutes. As a result of the differences between the two setups, these experiments are probing different types of stress fibre mechanical response. In contrast to severing experiments, where sarcomeres have been reported to collapse to a minimal length following a reduction in tension[68], we do not see changes in the spacing of the sarcomeric structure of crosslinkers in the fibre during the increase in tension driven by local RhoA activation. Our experiments thus probe the homeostasis of the system, while severing probes the limits of stability. Interestingly, it has been reported previously that sarcomere size in stress fibres fluctuates on a 20 min timescale[70], which might be related to the 50 min viscoelastic timescale revealed by our model results. Future insights could be derived from fluctuation analysis during optogenetic activation.

Contraction is driven by the local accumulation of myosin in the region of activation, but the origin of the elastic-like recovery is less clear. We see no evidence of additional myosin accumulation in distal portions of the stress fibre during the relaxation, suggesting that the recoil arises from zyxin-mediated repair mechanisms within stress fibres. This is further supported by the fact that the zyxin[(−/−)] cells exhibit identical contractile behaviour, but none of the elastic recoil. Zyxin is known to recruit actin polymerization factors such as VASP and actin crosslinkers such as alpha-actinin[59], and these interactions might be essential to ensure mechanical integrity of stress fibres. The molecular detail of these protein interactions under applied load remains an open question.

Our experiments highlight the importance of defining the relevant timescales and perturbations when describing a material as elastic or viscous. Given that typical turnover rates for proteins in the cytoskeleton are on the order of tens of seconds[15,25,26], it is surprising that our model results suggest that stress fibres have elastic-like properties on timescales of ∼1 h. The viscous behaviour of cells is typically associated with irreversible changes brought on through remodelling and dynamic activity of proteins (for example, cytoskeletal remodelling during migration)[21,71]. This has informed much of the active fluid theoretical frameworks that have been developed[21]. Conversely, elasticity has typically been used to describe cellular material properties without consideration of the dynamic activity of the components[44,72,73] and has been used in understanding force generation[44] and mechanosensing[74]. Our results suggest that active remodelling within the cytoskeleton can result in elastic-like properties of timescales exceeding those of dynamics of internal components. Moreover, our results indicate that very different viscoelastic timescales might co-exist in the same cell and even in the same cytoskeletal structures. Finally, our results implicate an important role for zyxin in regulating transitions between fluid-like and elastic-like behaviours. This has exciting implications for interpreting the underlying physics of active cytoskeletal materials. The fact that this behaviour can be controlled by the activity of a single protein suggests that there are a number of intriguing potential mechanisms cells can use to regulate their mechanical properties during morphodynamics and development.

## Methods

**Cell culture and transfection.** NIH 3T3 fibroblasts (American Type Culture Collection, Manassas, VA) were cultured in DMEM media (Mediatech, Herndon, VA) and supplemented with 10% foetal bovine serum (HyClone; ThermoFisher Scientific, Hampton, NH), 2 mM L-glutamine (Invitrogen, Caarlsbad, CA) and penicillin–streptomycin (Invitrogen). Zyxin$^{(-/-)}$ and zyxin$^{(-/-)}$+ EGFP-zyxin mouse embryonic fibroblast cells were a gift of Mary Beckerle's laboratory (University of Utah, Salt Lake City, UT) and cultured similarly to the NIH 3T3 fibroblasts[59]. Cells were tested for mycoplasma and were free of contamination. All cells were transiently transfected via electroporation 24 h before experiment using a Neon Transfection system (ThermoFisher Scientific). Following transfection, cells were plated on glass coverslips and imaged the next day.

**Drug treatments.** Cells were treated with either the 10 μM SMIFH2, a pan-formin inhibitor[38], or 1 μM of Y-27632, which inhibits ROCK (ThermoFisher Scientific), for at least 30 min before imaging.

**Plasmids.** The optogenetic membrane tether consisting of Stargazin-GFP-LOVpep and prGEF constructs used are previously described[6]. prGEF-YFP was constructed in an identical manner to prGEF with YFP replacing mCherry. This construct was used in experiments where the effects on various downstream markers were visualized. To examine effects on the actin and myosin networks, we used mApple-Actin and mApple-MLC constructs (gifts from M Davidson, University of Florida, Gainesville FL), mCherry-Vinculin (gift from V Weaver, University of California at San Francisco, San Francisco, CA) and mCherry-zyxin (gift from M Beckerle, University of Utah, Salt Lake City, UT).

**Live cell imaging.** Glass coverslips were placed in a Chamlide magnetic chamber (Live Cell Instrument, Seoul, Korea) in culture media supplemented with 10 mM HEPES and 30 μl ml$^{-1}$ Oxyrase (Oxyrase Inc., Mansfield, OH) and maintained at 37 °C. Cells were imaged on an inverted Nikon Ti-E microscope (Nikon, Melville, NY) with a Yokogawa CSU-X confocal scanhead (Yokogawa Electric, Tokyo, Japan) and laser merge module containing 491, 561 and 642 nm laser lines (Spectral Applied Research, Ontario, Canada). Images were collected on either a CoolSNAP HQ2 CCD (Roper Scientific, Trenton, NJ) or Zyla 4.2 sCMOS Camera (Andor, Belfast, UK). Local recruitment using the optogenetic probe was performed using a 405 nm laser coupled to a Mosaic digital micromirror device (Andor). Images were collected using a 60× 1.49 NA ApoTIRF oil immersion objective (Nikon). All hardware was controlled using the MetaMorph Automation and Image Analysis Software (Molecular Devices, Sunnyvale, CA).

Unless otherwise stated, cells were imaged in the 561 channel every 20 s for 45 min, with the first 15 min used to determine the steady state of the system, the second 15 min to perform local recruitment and the final 15 min to record any recovery. During recruitment, a local region drawn in MetaMorph was illuminated by the 405 nm laser for 960 ms at a power $< 1$ μJ s$^{-1}$ immediately before the acquisition of each 561 image.

**Local recruitment analysis.** All data analysis was performed using MATLAB (Mathworks, Natick, MA). Regions of interest were drawn to calculate the average intensity in the local recruitment region, a control area within the cell but far away from the recruitment area and a background area outside of the cell. The average background intensity was subtracted from the control region and this curve was used to determine a photobleaching correction. The photobleaching correction was then applied to the background subtracted average intensity in the local recruitment region and normalized to the average value of the first 15 min of the data.

**Focal adhesion analysis.** Images were thresholded and segmented to create binary masks using MATLAB. Adhesion masks were filtered to exclude adhesions $< 0.4$ μm$^2$ due to the inability to segment them consistently. The binary mask was then used to calculate the total number and average fluorescence intensity of adhesions in each frame. These masks were also used to calculate the average stress under the adhesions. To calculate the percentage of adhesions that increased in intensity or stress during activation, we compared the maximum intensity of the adhesion during activation to the intensity immediately before activation. Adhesions were considered to have shown an increase in either intensity or stress if the magnitude of the increase was $> 10$%.

**Kymograph and local displacement analysis.** Kymographs were created in MATLAB by drawing lines along stress fibres and averaging across a width of nine pixels. Local displacement was determined by locating a feature 5 μm from the edge of the activation zone in a kymograph immediately before activation. The location of this feature was then tracked and recorded following the 15 min period of activation and then again following 15 min of relaxation.

**Cytoskeletal flow analysis.** Images were first corrected for bleaching and then filtered with a 3D Gaussian filter to remove noise. Flow fields were calculated using an implementation of the optical flow algorithm by Brox et al.[49,50] that ensures spatial and temporal smoothness. Flow field kymographs were generated by projecting the flow vectors onto the line defining the kymograph. To compare the direction of flow with the organization of the cytoskeleton, the local orientation of actin fibres was extracted from the structure tensor[75].

**α-Actinin spacing analysis.** Kymographs were drawn as above. For each time point in the kymograph, local peaks in the linescan were determined. Peaks were then connected to create tracks across the kymograph. Local velocity was determined by isolating the section of the track during the activation period and fitting the trajectory to a straight line. The fitted slope was taken as the velocity.

**Traction force microscopy.** Traction force microscopy was performed as described previously[40,41,44]. Briefly, polyacrylamide gels embedded with 40-nm fluorescent microspheres (Invitrogen) were polymerized on activated glass coverslips. The shear modulus of the gels used in these experiments was 8.6 kPa. Following polymerization, gels were washed with PBS and crosslinked with the extracellular matrix protein fibronectin (Millipore, Billerica, MA) using the photoactivatable crosslinker sulfo-sanpah (Thermo Fisher Scientific). Cells were plated and allowed to spread for at least 4 h before imaging as described above.

Following imaging, cells were removed from the substrate using 0.5% SDS and a reference image of the fluorescent beads in the unstrained substrate was taken. The image stack was then aligned to correct for drift and compared to the reference image using particle imaging velocimetry to create a displacement field with a grid spacing of 0.86 μm. Displacement vectors were filtered and interpolated using the Kriging interpolation method. Traction stresses were reconstructed via Fourier Transform Traction Cytometry[40,76], with a regularization parameter chosen by minimizing the L2 curve[41]. The strain energy was calculated as one half the integral of the traction stress field dotted into the displacement field[44].

**Statistical analysis.** All experiments were repeated a minimum of three times. Cells presented in figures are representative samples of the population behaviour. Box plots represent the 25th, 50th and 75th percentiles of the data. Whiskers on the boxplot extend to the most extreme data points not considered outliers. Error bars represent the s.d., except where noted otherwise. Statistical significance was determined using independent two-sample Student's $t$-tests of the mean to compare groups of data. Statistical significance is indicated by asterisks: (*) represents a $P$ value $< 0.05$; (**) represents a $P$ value $< 0.01$.

**Code availability.** MATLAB analysis routines can be made available from the corresponding authors upon request.

**Data availability.** The data that support the findings of this study are available from the corresponding authors upon reasonable request. Stargazin-GFP-LOVpep and prGEF plasmids are available from Addgene.

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

## Acknowledgements

E.K.W. was supported by NIGMS T32 GM007183. C.A.B. and U.S.S. acknowledge support by the Heidelberg Karlsruhe Research Partnership (HEIKA). U.S.S. is a member of the cluster of excellence CellNetworks and the Interdisciplinary Center for Scientific Computing at Heidelberg. This work was also supported by NIGMS GM085087 to M.G. and NIGMS GM104032 to M.L.G.

## Author contributions

P.W.O, E.K.W, M.G. and M.L.G conceived the study and designed the experiments. P.W.O and E.K.W. performed experiments. P.W.O., D.P. and M.L. performed data analysis. E.K.W. designed and cloned the molecular constructs. C.A.B., D.P., M.L. and U.S.S. conceived and designed the theoretical model. C.A.B. and D.P. performed simulations. P.W.O., M.G., U.S.S. and M.L.G. wrote the manuscript with feedback from all authors.

## Additional information

**Competing interests:** The authors declare no competing financial interests.

