## [Peer Review File · Nature Communications]

Reviewers' comments:

Reviewer #1 (Remarks to the Author):

This is a well written and well done study. My suggestions are on improving the presentation of the current state of knowledge in the field and providing alternative interpretations of the work.

First, the paper does not discuss the experimentally measured mechanical model for the contraction of severed stress fibers in PMID 19751662 (see Figure 5 there). The elastic behavior of stress fibers deduced by the authors from their measurements is not consistent with the experimental observation that the elastic series element contributes minimally to the change in sarcomere length upon laser severing. That paper suggests that myosin contraction primarily determines sarcomere length; stress fiber length does not determine stress fiber tension. Moreover, the time scales here are a few seconds compared to the 10s of minutes over which the authors deduce elastic behavior.

Second, PMID 21225702 shows flow and fusion of stress fiber sarcomeres in living cells - similar to the report here with optogenetic activation (Figure 4) where speeds of flow are also reported. That paper should be discussed in the context of the authors' results and their interpretation of the backward flow as a result of elastic stresses in stress fibers. For example, the back flow could be interpreted as a result of the eventual decrease in actomyosin forces in the photoactivated spot resulting in a force imbalance favoring back flow (both actin and myosin return to baseline levels). PMID 21225702 and PMID 19751662 both discuss fluctuations in myosin activity and how that impacts flows and changes in stress fiber sarcomere lengths.

The authors suggest that the current understanding is that the actin cytoskeleton behaves like an elastic solid over time scales of typical 'kinetic processes'- I am not sure what this time scale is. But this statement should be revised in the light of the above papers- laser severing kinetics are very fast.

As the paper is titled 'harnessing optogenetics to probe cellular mechanics', the authors should cite another optogenetic study that activated Rac locally to increase local myosin tension, which causes motion of the nucleus PMID 24411232. It doesn't seem to me that the novelty here is in the use of optogenetics as a novel tool for subcellular mechanics.

- Tanmay Lele

Reviewer #2 (Remarks to the Author):

This paper uses an optogenetic technique to selectively (in both space and time) activate RhoA to probe its effect on stress fiber contractility. The two main results (aside from the technique) are of interest to both biological and physical scientists: (1) The important role of zyxin in preserving the elastic nature of the stress fibers which become fluid-like in the absence of zyxin. (2) The surprisingly long time scale of about 50 minutes during which the stress fibers respond to stress in an elastic manner (in cells that are not zyxin depleted). For these reasons, I think the paper will be of wide interest and recommend its publication in Nat. Comm. . I have the following suggestions that the authors may want to consider to clarify and perhaps strengthen the paper. Also, several

comments are aimed at widening the focus of the paper to address important issues in mechanobiology and their relation to the present results.

1. Page 3: end of first paragraph, a word is missing after ultimately - perhaps the word "are" before ultimately should be omitted.
2. Page 5: end of the middle paragraph "These results are consistent..." I suggest that authors be more explicit and say more explicitly what IS needed to drive changes in focal adhesion size. Based on their past work, it seems that stress fiber geometry near the adhesion templates the adhesion.
3. Are these results consistent with the experiments of Riveline et al. (not referenced in the manuscript) that showed that externally applied force causes focal adhesions to grow? Indeed, in that work there are stress fibers and is that the reason that the force caused growth?
4. Page 5: Bottom - estimates of mechanical work done by the cell. Is this the work done on the substrate immediately under the cell or does it also include the work done on the substrate relatively far from the cell? In computing this work from tractions, the question is whether the far-field tractions are included? Those are the tractions important for cell-cell interactions and not those directly beneath the cell.
5. Page 6: Top paragraph: "This suggests..." The authors may want to add that this is to be expected on very general grounds for force couples of equal and opposite forces arranged in a line. To a first approximation, the total force cancels within the line and the only remaining force is at the edges as they find experimentally.
6. Page 6: Contractile set point - first sentence. Tensional homeostasis was introduced long before Refs. 36-38 by Brown and Eastwood in quantitative measurements on cell stresses in *J. Cellular Physiology*, 1998 as can be seen by the title and content of their paper.
7. Contractile set point : This is important not only for the dynamical experiments described here but for cell response to other mechanical perturbations such as applied strain. Indeed, there is a controversy over whether it is passive elasticity (paper by Livne et al, *Nat. Comm.*, 2014) or active, stress-homeostasis (paper by De et al., *Nat. Phy.* 2007) that is responsible for the nearly perpendicular orientation of cells in applied, cyclic strain. The authors may therefore note that the set point is something of wide significance in mechanobiology.
8. Page 6: last paragraph "continuum model of the cell as a contactile element in series...". I think the authors mean that they model the cell, which is roughly a continuum, by a model that lumps the properties into one contractile element in series elastic and one viscoelastic element.
9. Page 7: Finding that stress fibers are elastic on time scales of about 50 minutes:
 - (a) What are the implications and relations of this to papers such as Ref. 21 on flowing, active gels? This might be important to sort out for the mechanobiology community where different groups use different conceptual approaches: cells as elastic objects vs. cells as flowing gels. Is the difference due to different focus on the stress fibers vs. cortex or is the difference due to well adhered cells vs. motile cells or to very different time scales considered?
 - (b) Either here or in the discussion, could the authors speculate about why the time scale for elastic response is so long?
10. Page 7: "myosin accumulated in the activation region.." The authors may want to read and reference a recent paper by Shiqiong Hu et al. in *Nat. Cell. Bio.* January, 2017 that discusses myosin flow (and even accumulation) along stress fibers.
11. Page 8: "embedded in a passive viscoelastic network": The authors may want to note that this is important for contractile forces mediated by this passive network that can cause various types of acto-myosin ordering: see the paper in point 10 above that images the disordered intervening actin network and suggests that this can transmit forces that locally order acto-myosin segments.

Reviewer #1:

This is a well written and well done study. My suggestions are on improving the presentation of the current state of knowledge in the field and providing alternative interpretations of the work.

We appreciate the reviewer's assessment that this is a "well written and well done study" and have taken the suggestions for improvement seriously. We have now added two paragraphs to the discussion, which we hope addresses these concerns. We hope the reviewer now finds this manuscript suitable for publication.

First, the paper does not discuss the experimentally measured mechanical model for the contraction of severed stress fibers in PMID 19751662 (see Figure 5 there). The elastic behavior of stress fibers deduced by the authors from their measurements is not consistent with the experimental observation that the elastic series element contributes minimally to the change in sarcomere length upon laser severing. That paper suggests that myosin contraction primarily determines sarcomere length; stress fiber length does not determine stress fiber tension. Moreover, the time scales here are a few seconds compared to the 10s of minutes over which the authors deduce elastic behavior.

We thank the reviewer for bringing up this important point. We have now included it in our citations, and agree that our experiments should be discussed in the context of the previous important findings that have been established via severing of stress fibers. We also agree that the stress fibers exhibit sarcomeric organization (elegantly verified in Hu et al. Nat Cell Biol 2017) and that their tension should not, therefore, be determined by their length. We also agree of a length-independent tension, which follows naturally from a physical force balance for a 1-dimensional system with sarcomeric organization.

As the reviewer suggests, there are important differences between our optogenetically induced contraction experiments and severing experiments. In a severing experiment, the mechanical stability of the fiber is compromised while the myosin activity remains unchanged. This leads to collapse of the stress fiber over second time scales. In our experiments, the myosin activity is elevated (and then only in the region of activation) and the stress fiber responds to elevated tension over tens of minutes. Thus, the severing is probing the response to catastrophic damage resulting in large contractile strains over seconds, whereas the optogenetic experiments is probing response to a small perturbation in tension, resulting in smaller strains over tens of minutes. The elastic component that we report here seems to arise from active remodeling due to the continuous repair activity of zyxin over a relatively long time scales. Although the stress fiber undergoes adaptation right after local changes in tension, the fiber is still fully intact and under tension during the whole course of our experiments. We now address with greater clarity these differences in a new paragraph in the discussion section of the manuscript (lines 306-317 in the manuscript).

Second, PMID 21225702 shows flow and fusion of stress fiber sarcomeres in living cells - similar to the report here with optogenetic activation (Figure 4) where speeds of flow are also reported. That paper should be discussed in the context of the authors' results and their interpretation of the backward flow as a result of elastic stresses in stress fibers. For example, the back flow could be interpreted as a result of the eventual decrease in actomyosin forces in the photoactivated spot resulting in a force imbalance favoring back flow (both actin and myosin return to baseline levels). PMID 21225702 and PMID 19751662 both discuss fluctuations in myosin activity and how that impacts flows and changes in stress fiber sarcomere lengths.

We thank the reviewer for the helpful comments. We completely agree that flow exists in stress fibers anyway, and that our activation essentially accelerates this flow. We now cite the mentioned references, which reports flow velocities that agree with our measurements.

We agree that changes in flow direction can arise from spatiotemporal variation in actomyosin forces along stress fibers. Indeed this is how we understand flow towards the region of activation-induced increases in myosin density. However, at the end of the activation, actomyosin in this region returns to baseline levels (Figure 1). This suggests that the backflow is not caused by diminished myosin activity, as further confirmed by our zyxin experiments (see below).

We agree that, in principle, there could be distal changes in actomyosin density that drive a backflow after activation ends. We do not observe this in our imaging of actin or myosin. However, since we cannot completely exclude this possibility, we have now added this possibility in the discussion section.

We have not performed a fluctuation analysis, but this would be an interesting avenue for future work. We note that the 20 min time scale mentioned in the fluctuation paper might be related to the 50 min time scale revealed by our model, as from a physics point of view, dissipation and fluctuations are strongly coupled.

The authors suggest that the current understanding is that the actin cytoskeleton behaves like an elastic solid over time scales of typical 'kinetic processes'- I am not sure what this time scale is. But this statement should be revised in the light of the above papers- laser severing kinetics are very fast.

We agree that this point should be explained more carefully and in greater detail. We have thus adapted our discussion accordingly. It is true that the dynamics and kinetics of the stress fiber following severing are quite rapid (i.e. less than a minute) and that this is driven mostly by rapid cycling of unloaded myosin motors. We believe these processes are very different from the homeostasis of stress fibers that we investigate here, when motors remain under load with slower kinetics.

The kinetic processes that we are referring to are the exchanges of proteins with those in the cytoskeleton. These types of kinetics are normally illuminated via FRAP experiments. A recent paper (Hu et al. Nat Cell Biol 2017) illustrated that myosin minifilaments within the sarcomere exchange on a timescale of ~60s, consistent to other works that we cited. One would thus expect perturbations on timescales only less than this exchange rate would behave elastically. It is therefore surprising that we see reversal of the flow (i.e. elastic behavior on a long timescale) because the rapid exchange of components would suggest that the system should have no memory of its prior state. The 50 min timescale reported in our manuscript follows from the fit to the mathematical model (ratio of viscosity and elastic modulus) and should be regarded as a statement on the macroscopic and steady state properties of the fibers. We note that the paper by Russell et al. in Cytoskeleton 2011 actually has reported a 20 min time scale for sarcomere length fluctuations, which might be related to our 50 min time scale following from the model. We have added additional text to the discussion to clarify this.

As the paper is titled 'harnessing optogenetics to probe cellular mechanics', the authors should cite another optogenetic study that activated Rac locally to increase local myosin tension, which causes motion of the nucleus PMID 24411232. It doesn't seem to me that the novelty here is in the use of optogenetics as a novel tool for subcellular mechanics.

We agree that the title is vague and have opted to change it to something more specific. We have now changed it to “Optogenetic Control of RhoA Reveals Zyxin-mediated Elasticity of Stress Fibers”. We also have adapted the abstract to the changed title and to the other changes made in the text. Additionally we have now included a citation of the mentioned reference. We also now include a reference to a recent paper by Valon et al. in Nature Communications titled “Optogenetic control of cellular forces and mechanotransduction” (published Feb 10 2017). With our new title, we also demonstrate that our work is focused on a detailed analysis of subcellular (stress fiber) dynamics, while the Valon et al. paper goes from single cells to cell collectives. We agree that there have been important earlier papers using optogenetics to control the cytoskeleton, and we now cite the Wu et al. paper in BPJ 2014 in the introduction.

Reviewer #2 (Remarks to the Author):

This paper uses an optogenetic technique to selectively (in both space and time) activate RhoA to probe its effect on stress fiber contractility. The two main results (aside from the technique) are of interest to both biological and physical scientists: (1) The important role of zyxin in preserving the elastic nature of the stress fibers which become fluid-like in the absence of zyxin. (2) The surprisingly long time scale of about 50 minutes during which the stress fibers respond to stress in an elastic manner (in cells that are not zyxin depleted). For these reasons, I think the paper will be of wide interest and recommend its publications in Nat. Comm. I have the following suggestions that the authors may want to consider to clarify and perhaps strengthen the paper. Also, several

comments are aimed at widening the focus of the paper to address important issues in mechanobiology and their relation to the present results.

1. Page 3: end of first paragraph, a word is missing after ultimately - perhaps the word "are" before ultimately should be omitted.

Thank you for catching this error. We have removed the “are” from the sentence.

2. Page 5: end of the middle paragraph "These results are consistent..." I suggest that authors be more explicit and say more explicitly what IS needed to drive changes in focal adhesion size. Based on their past work, it seems that stress fiber geometry near the adhesion templates the adhesion.

We thank the reviewer for this suggestion and agree that this is an important point to emphasize. We have now expanded this sentence to state that, as the adhesions and stress fibers are fully formed already in these cells, the optogenetic activation does not lead to *de novo* focal adhesion or stress fiber assembly, as shown in Fig. 2. The lack of change in adhesion morphology is consistent with previous results showing that adhesion size and tension are correlated mainly during the initial stages of assembly (Stricker et al Biophys J 2011), and arises from proximal stress fiber assembly rather than local tension per say (Oakes et al. J Cell Biol 2012).

3. Are these results consistent with the experiments of Riveline et al. (not referenced in the manuscript) that showed that externally applied force causes focal adhesions to grow? Indeed, in that work there are stress fibers and is that the reason that the force caused growth?

We first note that the Riveline experiment has been performed in serum-starved cells, such that focal adhesions and stress fibers could form *de novo*, while in our experiments, we already have a well-developed contractile system that is then stimulated to even larger contraction, see our response to the last comment. For a more general discussion of focal adhesion and stress fiber assembly, we refer to one of our recent reviews (Oakes et al. Curr Opin Cell Biol 2014). The timescales of the experiments in the Riveline paper are actually consistent with the timescales it takes to polymerize stress fibers (see Aratyn-Schaus et al. Mol Biol Cell 2011). We thus believe that the results presented here are consistent with the idea that tension doesn't promote adhesion growth and that the actin dynamics are a more important factor.

4. Page 5: Bottom - estimates of mechanical work done by the cell. Is this the work done on the substrate immediately under the cell or does it also include the work done on the substrate relatively far from the cell? In computing this work from tractions, the question is whether the far-field tractions are included? Those are the tractions important for cell-cell interactions and not those directly beneath the cell.

We calculate the mechanical work done by the cell as the energy needed to deform the substrate at the cell-substrate interface, because all deformations in this setup are caused by the traction forces. This is a method that is standard in the TFM field (see Butler et al. Am J Physiol. Cell Physiol 2002; Oakes et al Biophys J 2014). From the viewpoint of elasticity theory, this is exactly the same energy that is then stored in the substrate deformations, as can be shown using the Cauchy-Navier equation and the Gauss theorem for integrating the deformations (plus assuming that deformations vanish at infinity). In fact this is the same physics as pulling a harmonic spring open with a force that is always balancing the elastic restoring force: then the invested energy is exactly the one stored in the spring. What is neglected here are any viscous energies that might be dissipated during the process. We typically choose cells that are isolated from their neighbors so that we can ensure the deformation, and thus the mechanical energy, are the result of just the cell of interest.

5. Page 6: Top paragraph: "This suggests..." The authors may want to add that this is to be expected on very general grounds for force couples of equal and opposite forces arranged in a line. To a first approximation, the total force cancels within the line and the only remaining force is at the edges as they find experimentally.

The reviewer is correct and we have added a sentence to clarify this explicitly.

6. Page 6: Contractile set point - first sentence. Tensional homeostasis was introduced long before Refs. 36-38 by Brown and Eastwood in quantitative measurements on cell stresses in J. Cellular Physiology, 1998 as can be seen by the title and content of their paper.

We thank the reviewer for pointing this out to us. We have updated our citations (ref #45 in lines 120 and 293 of the manuscript) to properly reflect this.

7. Contractile set point: This is important not only for the dynamical experiments described here but for cell response to other mechanical perturbations such as applied strain. Indeed, there is a controversy over whether it is passive elasticity (paper by Livne et al, Nat. Comm., 2014) or active, stress-homeostasis (paper by De et al., Nat. Phy. 2007) that is responsible for the nearly perpendicular orientation of cells in applied, cyclic strain. The authors may therefore note that the set point is something of wide significance in mechanobiology.

We agree and have now included these citations and commented on this in the discussion. See lines 302-305 in the discussion section of the text.

8. Page 6: last paragraph "continuum model of the cell as a contractile element in series...". I think the authors mean that they model the cell, which is roughly a continuum, by a model that lumps the properties into one contractile element in series elastic and one viscoelastic element.

We agree that this wording is misleading and have changed it to “We constructed a model of the cell as a continuum of contractile elements in series. Each contractile element is in parallel with one elastic and one viscous element in series, thus making our model effectively an active Maxwell model for flowing material.” In order to make this point clear to the general reader, we have also changed the cartoons representing this model. For the expert, we note that there are different ways to schematize the model used here (compare Tlili et al. Eur. Phys. J. E 2015, Figure 7). We now choose a version which is more intuitive for the general reader. The same viscoelastic Maxwell model is typically used in active gel theory (Prost et al Nat Phys 2015), e.g. when laser cutting stress fibers (Saha et al Biophys J 2016).

9. Page 7: Finding that stress fibers are elastic on time scales of about 50 minutes: (a) What are the implications and relations of this to papers such as Ref. 21 on flowing, active gels? This might be important to sort out for the mechanobiology community where different groups use different conceptual approaches: cells as elastic objects vs. cells as flowing gels. Is the difference due to different focus on the stress fibers vs. cortex or is the difference due to well adhered cells vs. motile cells or to very different time scales considered?

We agree that this is an important point that deserves greater discussion and we have added a paragraph to cover these points in the discussion. From the viewpoint of continuum mechanics, biological systems in general are thought to be a viscoelastic-plastic. It depends on the time scale, the specifics of the system under consideration and the type of perturbation used, to determine whether it is appropriate to describe process as primarily viscous or elastic. While active fluid theory focuses on long time scales and flow phenomena, like cell migration, lamellipodial dynamics or cortical dynamics (see review by Prost et al in Nat Phys), there are other situations like mechanosensing, mechanical integrity of endothelial monolayers or wound healing, in which elastic aspects are more important (compare the review by Schwarz and Safran Rev Mod Phys 2013). We find it remarkable that here we start with a Maxwell model for a flowing system, and then find by comparison between experiment and model predictions that the ratio of viscous and elastic moduli depends on the molecular organization of the cell (presence or absence of zyxin) and still gives an order of magnitude that is above molecular exchange processes.

From the more molecular point of view, it is important if one considers the system as a whole or individual components. For instance, if one were just to look at the exchange kinetics of the proteins that comprise the stress fiber, one would be inclined to describe the system as viscous, as usually done for the cortex. The elastic behavior only reveals itself on a larger length scale and with an induced contraction along individual fibers. In our view, this is a nice example of emerging properties in a multiscale system.

Our experiments suggest that we have to examine active systems with great care regarding their details, as done here for local activation of stress fibers in well-adhered cells, and that the relation between microscopic and macroscopic behavior can be rather surprising. In particular, we note that very different viscoelastic relaxation times seem to exist in related systems (cortex with below minute time scales versus stress fibers with tens of minutes time scale), and thus one has to be careful when making statements on whole-cell mechanics, because it depends on the exact mode of probing which of the different time scales will dominate.

(b) Either here or in the discussion, could the authors speculate about why the time scale for elastic response is so long?

We are always happy to speculate on the caveat that further experimentation is surely needed to explore this interesting phenomenon. One hypothesis is that zyxin is somehow altering the structure of stress fiber at the interface to the adhesion (i.e. possibly allowing the actin to polymerize in a pre-strained state), such that when the contraction is released, the stress fiber can relax to its unperturbed state. Another possibility is that it regulates actin polymerization along the fiber, as suggested by the AFM-experiments in Colombelli et al. J Cell Sci 2009, which showed zyxin recruitment in a sarcomeric fashion to strained stress fibers. At any rate, zyxin is a mechanosensing protein that can repair mechanical failure and recruits molecules like VASP for polymerization and alpha-actinin for crosslinking. Future experiments are required to elucidate the molecular details of these repair processes.

10. Page 7: "myosin accumulated in the activation region" The authors may want to read and reference a recent paper by Shiqiong Hu et al. in Nat. Cell. Bio. January, 2017 that discusses myosin flow (and even accumulation) along stress fibers.

We agree with the reviewer and have now included multiple references to this work. It came out after our initial submission or we would have cited it sooner! We believe that our work is very consistent with their findings, especially regarding the flow and sarcomeric organization of myosin on the stress fibers.

11. Page 8: "embedded in a passive viscoelastic network": The authors may want to note that this is important for contractile forces mediated by this passive network that can cause various types of acto-myosin ordering: see the paper in point 10 above that images the disordered intervening actin network and suggests that this can transmit forces that locally order acto-myosin segments.

We agree and have added such a remark to our discussion section.

REVIEWERS' COMMENTS:

Reviewer #1 (Remarks to the Author):

I have no further comments.

Reviewer #2 (Remarks to the Author):

The authors have addressed all of my concerns. There are a few remaining small points that I suggest that they check. After they do so, the paper can be published in Nature Communications.

1. In point 4 on the work done by the cell, I suggest that the text clarify that this includes both the near field and far field deformations.
2. In point 7 on homeostasis: The authors mention this on lines 291 and 303 of the revised manuscript. But they do not clarify whether this homeostasis is consistent with activity (stress or active-strain) or with the passive elastic response of the cell to mechanical deformations -- which really should not be termed homeostasis since it exists for any elastic material. It would be of interest to know if their data supports active homeostasis (as first pointed out by Brown and Eastwood) or not (passive elastic response as in Ref. 65.).
3. Point 11: I could not find the mention of the imaging of the disordered actin network (Ref. 26) to support the idea mentioned in the present manuscript of the stress fibers as contractile elements in a passive viscoelastic network.

We appreciate the reviewers' feedback and have addressed their comments below.

Reviewer #1:

I have no further comments.

We are happy to learn that reviewer #1 recommends publication as is.

Reviewer #2:

The authors have addressed all of my concerns. There are a few remaining small points that I suggest that they check. After they do so, the paper can be published in Nature Communications.

We are happy to learn that reviewer #2 recommends publication after very minor changes.

1. In point 4 on the work done by the cell, I suggest that the text clarify that this includes both the near field and far field deformations.

We have now modified this sentence to include the phrase "...includes both near and far field deformations...". This change can be found on lines 106-107 of the revision.

2. In point 7 on homeostasis: The authors mention this on lines 291 and 303 of the revised manuscript. But they do not clarify whether this homeostasis is consistent with activity (stress or active-strain) or with the passive elastic response of the cell to mechanical deformations -- which really should not be termed homeostasis since it exists for any elastic material. It would be of interest to know if their data supports active homeostasis (as first pointed out by Brown and Eastwood) or not (passive elastic response as in Ref. 65.).

We appreciate the comments of the reviewer. We have added the following sentence (lines 304-306 in the revision) to clarify this: "Our optogenetic experiments also reveal a homeostatic mechanism, which depends on the repair protein zyxin. This suggests that the homeostasis we describe has evolved to be actively maintained and is not simply an elastic response of a passive material." We have further clarified that this is an active process in our system in line 291 by adding the word "actively" to our description.

3. Point 11: I could not find the mention of the imaging of the disordered actin network (Ref. 26) to support the idea mentioned in the present manuscript of the stress fibers as contractile elements in a passive viscoelastic network.

We have now added the following clause to line 185 of the revision: "...which is consistent with recently published image based observations²⁶".